# Multi-Scale Task Multiple Instance Learning for the Classification of Digital Pathology Images with Global Annotations

**First Author**                                            AUTHOR1.RESEARCH.EDU
*Anonymous*

**Second Author**                                           AUTHOR2.RESEARCH.EDU
*Anonymous*

**Editor:** Editor

## Abstract

Whole slide images (WSIs) are high-resolution digitized images of tissue samples, stored including different magnification levels. WSIs datasets often include only global annotations, available thanks to pathology reports. Global annotations refer to global findings in the high-resolution image and do not include information about the location of the regions of interest or the magnification levels used to identify a finding. This fact can limit the training of machine learning models, as WSIs are usually very large and each magnification level includes different information about the tissue. This paper presents a Multi-Scale Task Multiple Instance Learning (MuSTMIL) method, allowing to better exploit data paired with global labels and to combine contextual and detailed information identified at several magnification levels. The method is based on a multiple instance learning framework and on a multi-task network, that combines features from several magnification levels and produces multiple predictions (a global one and one for each magnification level involved). MuSTMIL is evaluated on colon cancer images, on binary and multilabel classification. MuSTMIL shows an improvement in performance in comparison to both single scale and another multi-scale multiple instance learning algorithm, demonstrating that MuSTMIL can help to better deal with global labels targeting full and multi-scale images.

**Keywords:** Multi-Scale Multiple Instance Learning, Multiple Instance Learning, Multi-scale approach, Computational pathology.

## 1. Introduction

Histopathology is the gold standard for diagnosing many diseases, such as cancer (Aeffner et al., 2017). Computational pathology involves the automatic analysis of digitized histopathology images, called whole slide images (WSIs). WSIs format includes several magnification levels of the samples, each one stored with a different spatial resolution. Each level allows visualizing different tissue patterns and morphologies (e.g. glands in low magnification levels (5x), single cells in the higher magnification levels (20x-40x)). Pathologists usually analyze the contextual information of the tissue at low magnification levels, identifying regions of interest and then zooming through them to analyze the tissue details and to confirm the disease findings at lower levels. The combination of contextual and details information, identified at several magnification levels, leads to the global diagnosis of the image.

Training machine learning algorithms for the automatic analysis of digital pathology images is still an open challenge (Cheplygina et al., 2019), also due to the limited availability of large datasets with local annotations and due to the multi-scale structure of the images.

Convolutional Neural Networks (CNNs) are currently the state-of-the-art for computational pathology tasks such as classification of WSIs (Jimenez-del Toro et al., 2017). CNNs usually require many locally (pixel-wise) annotated samples to train models effectively (Komura and Ishikawa, 2018). Local annotations are not always available, as their collection is an expensive and time-consuming process that usually requires the involvement of pathologists. Most publicly available datasets (Courtiol et al., 2018) do not include local annotations but many are paired with medical reports, which are inherently high-level text descriptions of the image content. Pathologists can analyze reports and extract information that can be used as a global (weak) label for the image. This kind of label refers to the whole image and does not include any information regarding the regions of interest used for performing the diagnosis and about the magnification levels used for the diagnosis (Karimi et al., 2020). CNNs do not easily handle the multi-scale structure of the WSIs, due to the fact that they are not scale-equivariant by design (Marcos et al., 2018). A scale-equivariant transformation is a transformation that, when the input is scaled of a factor $f$, produces an output scaled of a factor $f$ (Lenc and Vedaldi, 2015; Tensmeyer and Martinez, 2016). When a scale transformation is applied to CNN input data, its effect on the CNN output is unpredictable. Therefore, abnormalities must be identified in the proper range of magnification levels.

Recently, new methods were proposed in computational pathology to face the lack of local annotations (such as Multiple Instance Learning, MIL) and to face the lack of information about the magnification levels used (such as approaches to combine multi-scale images in CNNs' training), however, few studies target the combination of the two approaches. MIL (Hashimoto et al., 2020; Campanella et al., 2019; Lu et al., 2020; Mercan et al., 2017; Sudharshan et al., 2019; Wang et al., 2019) includes weakly-supervised algorithms that allow facing the lack of information about the regions of interest. WSI classification can be formulated as a MIL problem, where a WSI represents a bag $X_n$ that includes $P$ patches and the information available on the data concerns the entire WSI. Approaches to combine multi-scale images in CNN training can involve architectures where each magnification has its own branch to extract and combine features (Hashimoto et al., 2020; Jain and Massoud, 2020; Yang et al., 2019), U-Net based networks (Bozkurt et al., 2018; van Rijthoven et al., 2021) and CNNs where the convolution layers include multiple receptive fields (Li et al., 2019; Lai and Deng, 2017). These approaches allow to face the lack of information regarding the magnification levels involved in the diagnosis, combining contextual and detailed information identified at several magnification level. Few and only recent approaches combine MIL and multi-scale images, such as (Hashimoto et al., 2020), where the authors present a Multi-Scale Multiple Instance Learning (MSMIL) CNN.In this case, the CNN combines features from multi-scale patches in a MIL framework to obtain a global prediction for the WSI, showing a performance improvement over a CNN trained with patches from a single magnification level. However, the models present two main drawbacks: it does not provide outcomes at single magnification levels, different from what pathologists concretely do, and it requires several training phases, one for each of the single magnification levels and a training phase to combine the levels.

The MuSTMIL method described in this paper allows facing the lack of pixel-wise annotations and different spatial resolutions in CNN training, producing multiple predictions in a single training phase. MuSTMIL CNN has multiple scale branches as input (one for each magnification level) and produces multi-task predictions as output: one for each magnification level and a global prediction combining several levels. Differently from previous works (Hashimoto et al., 2020), the multiple outputs of the model allow to better optimize the entire model and take advantage of the combination of contextual and detailed information, since the global prediction influences and is influenced by the single-scale predictions like in a diagnostic process.

The method proposed in this paper is applied to the binary and multilabel classification of colon (colorectal) cancer, the fourth most commonly diagnosed cancer in the world (Benson et al., 2018). The diagnosis of the disease involves the detection of cancerous polyps (Ferlitsch et al., 2017), small agglomerations of cells, located on the colon border and the detection of glands. The visualization of low and medium magnifications allows to identify the glands. The dataset analyzed in this article includes WSIs with the corresponding global diagnosis. The diagnosis can include one or several colon tissue findings, among five classes: cancer, high-grade dysplasia (hgd), low-grade dysplasia (lgd), hyperplastic polyp and normal glands. The proposed MuSTMIL method outperforms both a Single-Scale Multiple Instance Learning (SSMIL) method and a baseline MSMIL method in binary and multilabel problems producing only global predictions in colon image classification.

**MULTI-SCALE TASK MULTIPLE INSTANCE LEARNING CNN**

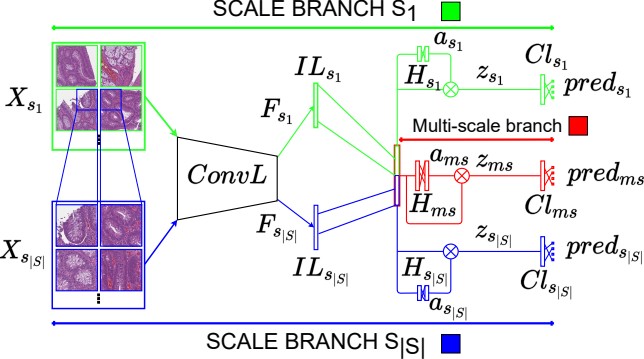

Figure 1: Overview of the MuSTMIL CNN. The magnification levels are noted as $s$, the combined magnification levels as $ms$. $X_s$ is a bag. ConvL is the convolutional layer block (shared among the branches). $F_s$ is the feature vector, $IL_s$ the intermediate fully-connected layer, $H_s$ the embedding vector, $z_s$ the output of the attention network. $Cl_s$ is the classifier, $pred_s$ the class prediction.

## 2. Methods

This paper proposes a MuSTMIL CNN that combine multi-scale images adopting a MIL framework to classify colon cancer WSIs. Figure 1 shows an overview of the CNN ar-

chitecture. The magnification levels are noted as $s \in S$ ($|S|$ representing the number of magnification levels adopted). The CNN includes multiple scale branches ($|S|$ branches, $\{s_1, \ldots, s_{|S|}\}$, one for each magnification level as input) and produces $|S|+1$ predictions ($|S|$ single-scale predictions and one multi-scale prediction) as output. Each scale branch receives as input a WSI $X_{ns}$, the corresponding label $Y_n$ and produces a prediction $pred_s$, for the corresponding magnification level $s$. Each scale branch includes convolutional layers, fully-connected layers, attention pooling layers and a classifier. The convolutional layers ($ConvL$) are used to extract the features ($F_s$). The fully-connected layers include an intermediate layer ($IL_s$), that produces smaller feature embeddings $H_s$ from $F_s$, composed of the patch embeddings $\{h_p\}_s$ ($p \in P$, $|P|$ representing the number of patches within a WSI). The attention pooling layer (Lu et al., 2020) aggregates the embeddings into a new array $z_s$, using an attention neural network ($w_s$ and $V_s$ are parameters of the network) that learns a function to weight ($a_s$ are the attention weights for each class) the embeddings and produces and aggregated embedding $z_s = a_s \otimes H_s$.

$$z_s = (\sum_{p=1}^{P} a_p h_{p_s}) \tag{1}$$

$$a_p = \frac{\exp(w_s^T \tanh(V_s h_{p_s}))}{\sum_{j=1}^{P} \exp(w_s^T \tanh(V_s h_{j_s}))} \tag{2}$$

The classifier receives input $z_s$ and outputs the class prediction ($pred_s$), for a fixed magnification level. Each branch is trained to optimize a Binary-Cross entropy loss function. The CNN also includes a multi-scale branch that produces a multi-scale prediction by aggregating features from several scale branches. Multi-scale concatenated embedding ($h_{ms} = h_0, h_1, \ldots h_S$) feeds the multi-scale branch and another attention network ($a_m s$ as attention weights), producing multi-scale aggregated embeddings $z_m s = a_m s \otimes h_{ms}$. The embeddings are used to feed a classifier ($Cl_m s$) that outputs the multi-scale global prediction $pred_{ms}$. The multi-scale branch is trained to optimize a loss function (binary-cross entropy). The optimization process of the network involves a loss function with multiple terms. The terms in the equation are the multi-scale loss function (weighted with $\alpha$) and the sum of the single-scale loss functions (weighted with $\beta$). This optimization leads to better performance also in the single-scale branches that benefit from the multi-scale features.

$$Loss = \alpha * Loss_{ms} + \beta * (\sum_{i=1}^{n} Loss_s) \tag{3}$$

## 3. Experiments

**Dataset** The MuSTMIL method is trained and evaluated on histopathology images of colon biopsies, polypectomies and tissue resections acquired during colonoscopy. Table 1 summarizes dataset composition. WSIs are provided from two medical hospitals and are acquired with ethics approval. The dataset includes over 2'000 WSIs, scanned with an Aperio and a 3DHistech scanners and stained with Hematoxylin and Eosin (H&E). All images include a global diagnosis of the images provided by a pathologist and a small

Table 1: Overview of the dataset. WSIs are collected from two medical hospitals and are split in training, validation and testing partitions. Each WSIs can be labeled with one or more of the five classes.

| PARTITION/CLASS | Cancer | HGD | LGD | Hyperplastic | Normal | #WSIs |
|:---:|:---:|:---:|:---:|:---:|:---:|:---:|
| training | 380 | 344 | 740 | 294 | 478 | 1826 |
| validation | 59 | 48 | 132 | 40 | 88 | 305 |
| testing | 85 | 48 | 65 | 26 | 0 | 192 |
| **total** | **524** | **440** | **937** | **360** | **566** | **2323** |

subset comes with pixel-wise annotations, used to compare CNN predictions. The diagnosis includes one or more classes among: cancer, high-grade dysplasia, low-grade dysplasia, hyperplastic polyp and normal glands. The WSIs are analyzed at 5-10x magnification, since pathologists recognize these classes at low to medium magnifications.The dataset is split into three partitions: training (1'826 WSIs), validation (305 WSIs) and testing (192 WSIs pixel-wise annotated), so that all images from a patient are included in the same partition.

**Pre-processing** The image pre-processing involves the image splitting into a multi-scale bag $X_{ns}$ including several $X_s$ bags of patches, for each of the magnification levels involved. WSIs are split into a grid of patches $X_s$ for each magnification level $s$, starting from the highest magnification level available $M_m$. Considering that the CNN input layer needs patches of 224x224 in size ($p$), after the extraction patches are resized. Therefore, the size of the grid ($p_s$) varies depending on the magnification level $s$, as follows:

$$p : s = p_s : M_m \tag{4}$$

Patches coming from the same region are linked across magnification levels: the $i$-th patch from bag $X_1$, at lower magnification, includes the $j$-th patch from bag $X_2$ within the bag at higher magnification. Considering that bags with patches from lower magnification include fewer patches than bags with patches from higher magnification, the $i$-th patch at lower magnification can be linked with more patches at higher magnification level.

**Experimental setup** MuSTMIL, SSMIL and MSMIL CNNs have the same backbone architecture and are trained multiple times using the same strategy to set the hyperparameters to avoid overfitting and to face the class imbalance. The backbone architecture is a ResNet34 (pre-trained on ImageNet), used as a feature extractor (frozen during the training). It produces feature vectors of size 512 for each input patch. Each model is trained five times to limit the non-deterministic effect of the stochastic gradient descent used to optimize the model using the chosen hyperparameters. The average and standard deviation of the models are reported. The hyperparameters are chosen with a grid search (Chicco, 2017), aimed at finding the optimal configuration of the CNN hyperparameters (i.e. the configuration that allows the CNN to have the lowest loss function on the validation partition data). The hyperparameters involved in the grid search are the number of epochs

(five epochs), the optimizer (Adam), the learning rate ($10^{-3}$), the decay rate ($10^{-4}$), the number of nodes within the intermediate fully-connected layers (128) and the value of $\alpha$ and $\beta$ of the loss function ($\alpha=1$ and $\beta=1$). Overfitting and class imbalance are limited adopting a class-wise data augmentation method that uses three operations: rotations, flipping and colour augmentation. The augmentation is implemented with the Albumentations library (Buslaev et al., 2018).

## 4. Results

MuSTMIL outperforms a Single-Scale MIL (SSMIL) and a baseline Multi-Scale MIL method (MSMIL), on a binary and a multilabel classification problems. SSMIL is a CNN with the same backbone, trained with patches from a single magnification level. Baseline MSMIL CNN is based on Hashimoto et al. (2020) and produces only a global WSI prediction. Single-scale branches of the baseline method are trained with patches from a single level and then are combined; in order to guarantee a better comparison with the MuSTMIL proposed in this paper, the implementation of the method includes colour augmentation instead of the domain adversarial network proposed by the authors to address colour variability. Table 2 summarizes the results for all the methods.

Table 2: Performance of MuSTMIL CNN, for the binary (left) and the multilabel (right) problems. SSMIL is the Single-Scale Multiple Instance Learning CNN, Hashimoto et al. (2020) MSMIL is the baseline Multi-Scale Multiple Instance Learning and MuSTMIL the Multi-Scale Task Multiple Instance Learning CNN. MuSTMIL global prediction is the global output, while MuSTMIL $s$x branch is the output of the single scale branches of our CNN. Networks performance are assessed using accuracy and F1-score.

| MAGNIFICATION | binary problem | | multilabel problem | |
|---|---|---|---|---|
| | accuracy | F1-score | micro-accuracy | micro F1-score |
| SSMIL 5x | $0.836 \pm 0.022$ | $0.860 \pm 0.022$ | $0.815 \pm 0.025$ | $0.587 \pm 0.064$ |
| SSMIL 10x | $0.832 \pm 0.028$ | $0.866 \pm 0.029$ | $0.824 \pm 0.025$ | $0.597 \pm 0.076$ |
| Hashimoto et al. (2020) MSMIL | $0.849 \pm 0.022$ | $0.876 \pm 0.025$ | $0.840 \pm 0.015$ | $0.673 \pm 0.018$ |
| MuSTMIL global prediction | $\mathbf{0.870 \pm 0.011}$ | $0.893 \pm 0.010$ | $0.857 \pm 0.006$ | $0.682 \pm 0.008$ |
| MuSTMIL 5x prediction | $0.868 \pm 0.010$ | $\mathbf{0.892 \pm 0.009}$ | $\mathbf{0.863 \pm 0.009}$ | $\mathbf{0.683 \pm 0.015}$ |
| MuSTMIL 10x prediction | $0.857 \pm 0.018$ | $0.866 \pm 0.027$ | $0.855 \pm 0.020$ | $0.680 \pm 0.038$ |

The binary problem involves the classification of high-risk classes (cancer and high-grade dysplasia) and low-risk classes (low-grade dysplasia, hyperplastic polyps and normal glands). The performance is evaluated using accuracy and F1-score. MuSTMIL outperforms the SSMIL and the baseline MSMIL, considering both the global-branch and each of the single-scale branches. The multi-scale branch and the single-scale branch trained with patches from 5x reaches the highest performance.

The multilabel problem involves the classification of the five classes: cancer, high-grade dysplasia, low-grade dysplasia, hyperplastic polyps and normal glands. The test partition includes only malignant images, therefore there are no images labeled as normal glands. The performance is evaluated using micro-accuracy and micro F1-score. MuSTMIL outperforms

the SSMIL and the baseline MSMIL, considering both the global-branch and each of the single-scale branches. The single-scale branch trained with patches from 5x reaches the highest performance.

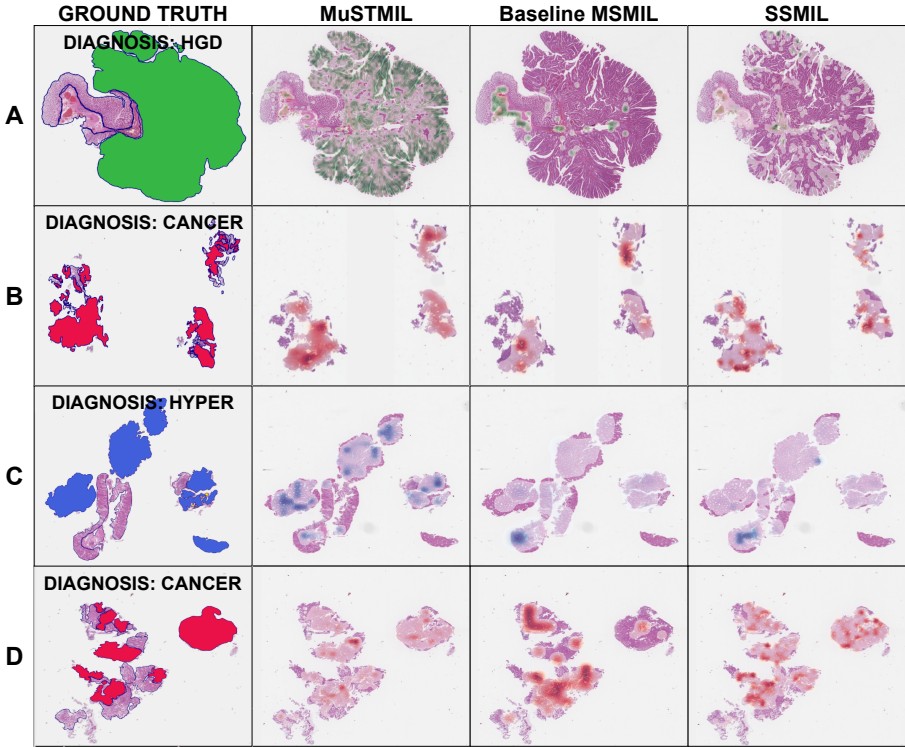

Figure 2: Comparison between pixel-wise annotations made by a pathologist with attention maps of MuSTMIL, Hashimoto et al. (2020) MSMIL and SSMIL compared: cancer (red), hgd (green), lgd (yellow), hyperplastic polyp (blue), normal tissue (orange). In rows 1-3, MuSTMIL obtains results qualitatively better than the other methods, while in the last row MuSTMIL does not fully highlight the relevant areas.

## 5. Discussion

The results obtained show that MuSTMIL CNN benefits of the multiple scales for training and the multi-task optimization of the CNN weights, obtaining higher performance compared with a SSMIL and a baseline MSMIL (Hashimoto et al., 2020) producing only a global prediction. Combining images from several magnification levels allows the model to focus on different details and combine both contextual and detailed information leading to the diagnosis. Figure 2 shows pixel-wise annotations made by a pathologist and attention heatmaps of MuSTMIL, baseline MSMIL and SSMIL in multilabel problem.In the top three rows, the attention maps produced by MuSTMIL better correspond to the pixel-wise annotations, while in the last row the baseline MSMIL and SSMIL produce better attention

maps. With multi-scale images as input and multiple predictions as output the models produce attentions maps focused on larger portions of the images, as shown in column MuSTMIL of Figure 2. This can be explained considering that the multi-scale input images and the training optimization of MuSTMIL allow the model to have feature representation including contextual and detailed information from different magnification levels. In the proposed MuSTMIL method, a multi-task loss function is optimized including a multi-scale loss function and a loss function for each magnification level. In this way, updates of the parameters within a single-scale branch are influenced not only by the backpropagation of the branch, but also by the other branches, since the features are combined in the multi-scale branch. Thus, the gradients are backpropagated into both the multi-scale and the single-scale branches, influencing the predictions and the branch attention weights. The results obtained show that, for the binary and multilabel classification tasks, MuSTMIL CNN outperforms SSMIL and MSMIL CNNs for both accuracy and F1-score. The accuracy metric performance means that the model produces more accurate predictions. The F1-score metric (combination of recall and precision) performance means that the model produces a better combination of false negatives (recall) and false positives (precision). This can be qualitatively understood by looking at the attention heatmaps in Figure 2. Both SSMIL and the baseline MSMIL produce attention heatmaps focused on small regions, while the attention map of MuSTMIL focuses on larger regions. The SSMIL and the baseline MSMIL suffer of the opposite problem, since they are more conservative in the attention, focusing usually only on small regions and then producing more false negatives. The MuSTMIL method proposed in this paper is more efficient by a computational point of view than the baseline MSMIL, since it requires only one phase to combine $n+1$ magnification levels, while the other method requires $n+1$ training phases (one training for each of the scales involved and a training phase to combine the branches).

## 6. Conclusions

This paper introduces a novel Multi-Scale Task Multiple Instance Learning (MuSTMIL) CNN to classify WSIs. The approach allows combining contextual and detailed information from multiple magnification levels, it has multiple scale branches as input and produces multiple single-scale and one multi-scale prediction. MuSTMIL outperforms a in binary and in multilabel colon WSI classification a SSMIL CNN and one of the multi scale multiple learning CNNs (Hashimoto et al., 2020) presented in literature for digital pathology, that produce only one global prediction. We plan to test MuSTMIL on additional data, other organs and with a larger number of scales. The code with the model pre-processing and implementation will be made publicly available on Github upon publication, benefiting the scientific community and allowing to reproduce the experiments.

## Acknowledgments

This project has received funding from the XXXXXX research and innovation programme under grant agreement No XXXXXXXXX.

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
