# OpenReview forum: "Multi-Scale Task Multiple Instance Learning for the Classification of Digital Pathology Images with Global Annotations"
_MICCAI.org/2021/Workshop/COMPAY — COMPAY 2021_

### Official Review · Reviewer_P7w8 · 2021-08-20
**New multi-scale approach applied to DP**

**Rating:** 8
**Confidence:** 3

**Review:**

This is a great paper and a clear accept from my point of view. In this work, the authors extend the repertoire of existing multi-scale approaches by introducing a "multi-scale task multiple instance learning" (MuSTMIL) method. The main difference with other approaches lie in the combination in data from all magnification levels in the feature extraction layer. This way, the extracted features inherently use multiple scales instead of having different feature extraction layers for each level. The authors demonstrate that this method works well to predict i) binary severity of colon polyps and ii) multi-label outcomes of colon polyps.

The paper is well written. However, it is too succinct to be able to reproduce the work. My main question deals with how patches from different levels are linked. The most obvious would be to break down a patch from a low-level magnification down in B^2 patches at a B times higher magnification. I.e., to have concentric patches from different magnifications aligned around a fixed center. But it appears this is not what is done in this paper. The authors' two statements "Considering that the CNN input layer needs patches of 224x224 in size (p), after the extraction patches are resized. Therefore, the size of the grid (ps) varies depending on the magnification levels" and "Considering that bags with patches from lower magnification include fewer patches than bags with patches from higher magnification, the i-th patch at lower magnification can be linked with more patches at higher magnification level." does not provide clarity on this important issue.

Minor issues:
- The description of the loss functions is incomplete: The authors only mention binary-cross entropy in the methods, but do report multi-class results as well.
- In figure 2 it is unclear which magnification level was used to draw the attention maps
- A more in-depth discussion about the size of the attention maps generated from MuSTMIL would be interesting: Would it be better to have larger/smaller maps for explainability? (Or put differently, is high the specificity achieved with the other methods sufficient to classify a slide?)

---

### Official Review · Reviewer_J21L · 2021-08-24
**Multi-scale Multi-task Multiple Instance Learning CNN for colorectal cancer**

**Rating:** 6
**Confidence:** 3

**Review:**

The paper Multi-scale Task Multiple Instance Learning for the Classification of Digital Pathology Images with Global Annotations proposes a CNN architecture that combines Multiple Instance Learning (MIL) and Multi-task Learning to detect colorectal cancer in Whole Slide Images.
The problem consists of predicting the grade of the global WSI  from five categories and on detecting cancer regions in the image.
The authors claim that by combining multi-task learning with multiple instance training the model may benefit of the extra information and thus improve the performance on both tasks. A major limitation of this paper is the lack of clarity about how the data splits were built. Why there are no negative samples in the testing set? One patient from the validation set could have been moved to the test set, for example, to have a broader evaluation. I feel that, in this way, the evaluation is a bit incomplete.
Below some pros and cons about this paper.
Pros:
- The paper is clearly written and easy to follow. There are a few typos and errors in the English grammar that should be addressed.
- The idea behind the paper is legitimate, although not very novel (only incremental)
- The qualitative results show better attention of the model than SingleScale MIL.

Cons:
- The architecture proposed here is very close to existing work (Hashimoto et al., 2020) and the novelty introduced by the paper is only incremental. With respect to the multi-scale multiple instance learning paper, the authors only add the loss of the individual branches to the main task loss. Notwithstanding, it is legitimate to verify whether this may not be a better strategy than that proposed by Hashimoto et al., 2020.
- I do not understand what decisions were made to generate the testing, validation and training splits. How were the patients assigned to each split? Why was the split not done in a stratified way ? This is a major limitation, which affects the evaluation, as also mentioned by the authors in the results section (Sec.4)
- The authors refer to "several" magnification levels in the paper, although they only use magnifications at 5x and 10x. Their results, besides, show that the magnification at 5x is sufficient to get the best results. I would encourage the authors to clarify this aspect to verify whether the additional 10x task is actually beneficial.

---

### Official Review · Reviewer_zh8D · 2021-08-25
**Multi-Scale Task Multiple Instance Learning**

**Rating:** 5
**Confidence:** 5

**Review:**

Multi-scale and the limitations of image-level annotation have always been the challenges for pathological image analysis. The author proposed a MuSTMIL method by multiple instance learning and multi-scale tasks. The experimental results of binary and multilabel colon cancer WSI classification show the capabilities of the proposed model.

However, following concerns are required to be further addressed:

1.	It is not difficult to understand that the introduction of images with multiple magnifications for training a robust model. The proposed method outperforms the baseline on binary and multilabel issues, however, the ablation experiment for each proposed module should be added and demonstrate clear contributions.

2.	This paper is overall challenging to follow due to wording and logical structure, especially for Section 2.


3.	What is the number of multiple scale branches |S|? Multi-scale is the key concern of this paper. It’s better to state clearly the principles of setting the number of branches and do some comparative experiments.

4.	The author mentioned “produces a multi-scale prediction by aggregating features from several scale branches”, the way of feature fusion affects the performance of multi-scale branch, so what is the specific operation process of "aggregating"?

5.	During training, Losss is used to guide multiple scale branches and Lossms is used to guide to guide the multi-scale branches of aggregating multiple scale features, it is questionable that they are repetitive and redundant, and please explain their different roles and the principle of parameter setting (α and β) in formula (3).

6.	According to Table 1, it’s noticeable that the test set does not contain the fifth type of sample (Normal), why? It may cause inaccurate validation of the model.

7.	The author introduces the attention model in each branch, please explain the motivation and implementation details.

8.	Which branch’s output is used to measure MuSTMIL and whether cross-validation is used?

9.	Looking at Figure 2, the five types of WSI prediction heatmaps should all be displayed for unbiased evaluation. In addition, please explain the reason of “while in the last row the baseline MSMIL and SSMIL produce better attention” mentioned in Discussion.

10.	For the comparison results, current version is short of comparison or discussion with state-of-the-art methods, such as following for reference:

Ilse M, Tomczak J, Welling M. Attention-based deep multiple instance learning. InInternational conference on machine learning 2018 Jul 3 (pp. 2127-2136). PMLR.
Hou L, Samaras D, Kurc TM, Gao Y, Davis JE, Saltz JH. Patch-based convolutional neural network for whole slide tissue image classification. InProceedings of the IEEE conference on computer vision and pattern recognition 2016 (pp. 2424-2433).
Wang X, Tang F, Chen H, Luo L, Tang Z, Ran AR, Cheung CY, Heng PA. UD-MIL: uncertainty-driven deep multiple instance learning for OCT image classification. IEEE journal of biomedical and health informatics. 2020 Mar 30;24(12):3431-42.

---

### Decision · Program_Chairs · 2021-08-25

Accept